# Crosstalk between Ca^2+^ Signaling and Cancer Stemness: The Link to Cisplatin Resistance

**DOI:** 10.3390/ijms231810687

**Published:** 2022-09-14

**Authors:** Sana Kouba, Frédéric Hague, Ahmed Ahidouch, Halima Ouadid-Ahidouch

**Affiliations:** 1Laboratoire de Physiologie Cellulaire et Moléculaire, Université de Picardie Jules Verne, UFR des Sciences, 33 Rue St Leu, 80039 Amiens, France; 2Département de Biologie, Faculté des Sciences, Université Ibn Zohr, Agadir 81016, Morocco

**Keywords:** calcium signaling, calcium channels, cisplatin, cancer stem cells, chemoresistance

## Abstract

In the fight against cancer, therapeutic strategies using cisplatin are severely limited by the appearance of a resistant phenotype. While cisplatin is usually efficient at the beginning of the treatment, several patients endure resistance to this agent and face relapse. One of the reasons for this resistant phenotype is the emergence of a cell subpopulation known as cancer stem cells (CSCs). Due to their quiescent phenotype and self-renewal abilities, these cells have recently been recognized as a crucial field of investigation in cancer and treatment resistance. Changes in intracellular calcium (Ca^2+^) through Ca^2+^ channel activity are essential for many cellular processes such as proliferation, migration, differentiation, and survival in various cell types. It is now proved that altered Ca^2+^ signaling is a hallmark of cancer, and several Ca^2+^ channels have been linked to CSC functions and therapy resistance. Moreover, cisplatin was shown to interfere with Ca^2+^ homeostasis; thus, it is considered likely that cisplatin-induced aberrant Ca^2+^ signaling is linked to CSCs biology and, therefore, therapy failure. The molecular signature defining the resistant phenotype varies between tumors, and the number of resistance mechanisms activated in response to a range of pressures dictates the global degree of cisplatin resistance. However, if we can understand the molecular mechanisms linking Ca^2+^ to cisplatin-induced resistance and CSC behaviors, alternative and novel therapeutic strategies could be considered. In this review, we examine how cisplatin interferes with Ca^2+^ homeostasis in tumor cells. We also summarize how cisplatin induces CSC markers in cancer. Finally, we highlight the role of Ca^2+^ in cancer stemness and focus on how they are involved in cisplatin-induced resistance through the increase of cancer stem cell populations and via specific pathways.

## 1. Introduction

Cancer is a group of diseases characterized by abnormal cell proliferation and high invasion potential to other parts of the body. At this point, several strategies have been developed for cancer treatment, including surgery, radiotherapy, chemotherapy, or targeted therapy [1,2]. Platinum complexes are used clinically as adjuvant therapy for cancers with the aim of inducing tumor cell death. Cisplatin was first synthesized in 1845 and, approximately 50 years later, its structure was established by Alfred Werner [3]. In the 1960s, several platinum-based compounds were tested in mice against the Sarcoma 180 and the Leukemia L1210 tumors. Among these compounds, cisplatin was the most efficient agent in reducing those tumors, with a 60–100% success rate [4]. These first test results supported the role of cisplatin as an anti-cancer drug and opened a new era in cancer treatment. Cisplatin induces cytotoxicity by interfering with transcription and/or DNA replication processes and damages tumors via apoptosis, a cell-death mechanism activated by several signal transduction pathways [5]. While a combination chemotherapy with cisplatin is essential for the treatment of various cancers, challenges occur because cancer cells could become cisplatin-resistant, which leads to treatment failure and cancer relapse. These collapses can be explained by the presence of resistant cell populations known as cancer stems cells (CSCs) [6]. CSCs can cause resistance because of certain properties that give them self-renewal abilities and differentiation potentials [7,8,9]. These CSC features are controlled by many intracellular factors, such as transcription factors or signaling pathways, as well as extracellular factors that are present in the tumoral microenvironment. It is now clear that among the dysregulated mechanisms in cancer, those related to calcium (Ca^2+^) signaling play significant roles in numerous aspects of this disease [10,11,12,13]. The nature of Ca^2+^ signals and their magnitudes are strictly regulated and are fundamental for cellular functions such as gene expression, proliferation, apoptosis, metabolism, and survival. Harmonious Ca^2+^ signaling is ensured by a wide variety of ion channels, pumps, and exchangers, coordinating together to guarantee the delicate balance between Ca^2+^ entry and release [14,15]. Altered Ca^2+^ signaling that is associated with an aberrant channel expression and/or function has been widely described in cancer; more recently, increased evidence based on research studies has shown that ion channels, including those responsible for Ca^2+^ transport, play a key role in CSC functions [16,17]. To complicate things further, cisplatin was shown to interfere with Ca^2+^ homeostasis [18,19,20,21]; with that being published and demonstrated by research, it is not surprising that cisplatin-induced aberrant Ca^2+^ signaling is linked to CSC behaviors leading to the emergence of a resistant phenotype. This review will be divided into three different sections. In the first section, we briefly summarize the mechanism of action of cisplatin and highlight the interference of cisplatin with Ca^2+^ homeostasis in tumor cells. In section two, we detail how cisplatin induces CSC markers in different types of cancer. In the final sections, we discuss how the different Ca^2+^ channels participate in cancer stemness and focus on how cisplatin treatment induces CSCs enrichment via Ca^2+^-dependent signaling pathways, thereby leading to resistance.

## 2. Cisplatin Action Mechanisms, in Brief, the Transduction Pathways Involved, and Ca^2+^-Related Resistance Mechanisms in Cancer Cells

### 2.1. Cisplatin Action Mechanisms in Brief

Cisplatin is a neutral, inorganic, square planar complex that reacts with DNA to induce either the repair of DNA damage and survival or the activation of an irreversible apoptotic program. The cytotoxicity of cisplatin is mainly attributed to its interaction with the purine-N(7) of guanine and adenine residues on DNA, mainly forming 1,2-intrastrand crosslinks between two adjacent guanines (1,2-d(GpG), ~65%) or the neighboring adenine and guanine (1,2-d(ApG), ~25%), as well as inter-strand crosslinks and mono-functional adduct. Cisplatin-induced DNA damage activates a number of proteins responsible for apoptotic or DNA repair pathways (Figure 1). In fact, the failure of an adequate repair results in aberrant mitosis of the cells, followed by apoptosis. Apoptosis is mediated not only by proteins such as the Bcl-2 family of proteins and p53-tumor suppressors but also by intracellular signal-transduction pathways that are often facilitated by protein kinases and phosphatidylinositol 3-kinase (Figure 1). At the molecular level, apoptosis is accomplished through caspase activation. Activated caspases target and invade both the nuclear and the cytoplasmic factors that are responsible for the maintenance of the architecture of the cell and participate in the repair of DNA, replication, and transcription. This process also depends on anti-apoptotic as well as apoptotic proteins (Figure 1). The action mechanism of cisplatin has been widely described in the literature. Therefore, in the current review, we will only represent an illustration that sums up the previously published effects of cisplatin (Figure 1). For further information regarding the detailed action mechanism of cisplatin and its related effects, we recommend that the readers refer to these reviews [3,5,22].

### 2.2. Link between Cisplatin Resistance and Ca^2+^ Homeostasis in Cancer Cells 

To begin with, it is now clear that the main reason for cancer onset is because of mutations in the oncogenes or tumor suppressor genes, which are directly involved in cell proliferation or death. In the same way, some genes have been identified to be the main contributors to cisplatin resistance, such as BRCA1 and BRCA2 genes. For example, it has been shown that acquired cisplatin resistance can be reversed by BRCA2 inhibition and that the anti-proliferative effects of cisplatin can be enhanced after BRCA1/2 inhibition as well [23,24,25]. This also prevents the formation of tumor metastases in vivo after cisplatin treatment. However, it is important to highlight that some other genes may be affected and may contribute to the switch to a more resistant phenotype; among these genes, those that encode calcium channels are present. Ca^2+^ homeostasis refers to the maintenance of a constant concentration of Ca^2+^ cations in the cell and its different compartments. This maintenance depends on fluxes from/to the extracellular compartment and Ca^2+^ transfer within organelles (Figure 2). Inter-organellar communication often takes the form of Ca^2+^ signals, which originate from the ER and target the mitochondria. The ER–mitochondrial Ca^2+^ transfer is possible through ER structures that are in close proximity to mitochondria and are known as the mitochondria-associated ER membranes or MAMs. These structures create micro-domains in which Ca^2+^ concentrations are higher than in the cytosol, allowing rapid mitochondrial Ca^2+^ uptake. It is well established that mitochondrial Ca^2+^ overload results in apoptosis; many chemotherapeutics depend on efficient ER–mitochondrial Ca^2+^ signaling to exert their function. However, several oncogenes and tumor suppressors present in the MAMs can alter Ca^2+^ signaling in cancer cells, rendering chemotherapeutics ineffective [26,27]. Despite the important role of MAMs in Ca^2+^ homeostasis and chemotherapy efficiency, we will not expand upon their implication in this process; the present review will be focusing on plasma membrane (PM) ion channels.

Fluctuations in the intracellular Ca^2+^ mediated by PM channels are known to activate a number of Ca^2+^-dependent pathways and effectors that control cellular processes, under both normal and oncogenic circumstances. Some of these Ca^2+^-sensitive effectors include proteins such as calmodulin (CaM), kinases, phosphatases, and transcription factors, of which, NF-κB, CREB (cAMP response element), and NFAT (nuclear factor of activated T) (Figure 2) were recently associated with proliferation, migration, and resistance in many cancer types [11,28,29]. The expression of Ca^2+^ channels has been demonstrated to be either upregulated or downregulated in various cancers and both phenomena have been linked to cancer progression and survival. Since Ca^2+^ disruption is associated with oncogenic transformation, its role as a potential target in anti-cancer therapies has been accentuated. As for chemoresistance, despite intensive studies focusing on the role of Ca^2+^ in this process, fragmentary and, sometimes, contradictory data are available regarding Ca^2+^ in cisplatin-dependent cytotoxicity. To understand how cisplatin interferes with Ca^2+^ homeostasis, we questioned whether direct interactions could occur between PM channels and the platinum-based compound. However, to our knowledge, cisplatin does not interact with PM ion channels but can form complexes with PM phospholipids [30]; this could lead to changes in the biophysical properties of the membranes (e.g., fluidity and permeability) which impacts the activity of some Ca^2+^ channels that are sensitive to lipids. Taken together, this process might affect the sensitivity and resistance phenotypes of cells to cisplatin. The importance of Ca^2+^ homeostasis in cisplatin effects (Table 1) in cancer was shown for the first time by Tachikawa et al., who found a sustained increase in intracellular Ca^2+^ concentration [Ca^2+^]_i_ in response to cisplatin in the KB epidermoid carcinoma cell line [31], suggesting the involvement of Ca^2+^ homeostasis alterations in cisplatin-resistant tumor cells. A couple of years later, another group measured [Ca^2+^]_i_ using imaging techniques, and found that cisplatin-resistant lung adenocarcinoma A549 cells (A549/DDP) exhibit less [Ca^2+^]_i_ (−33%) than do cisplatin-sensitive A549 cells [32]. Ever since this study, several studies have found that cisplatin increases [Ca^2+^]_i_ in several cancer in vitro models, including breast, ovarian, cervical, bladder, and neuroblastoma cell lines, leading to apoptosis [33,34,35,36,37,38]. However, the effectors involved in this increase remained unknown. The first studies using the 2-APB (2-Aminoethoxydiphenyl borate) pharmacological agent reported the involvement of IP_3_ receptors (IP3R) in cisplatin-induced ER Ca^2+^ release [18,34,36]. The implication of IP_3_R in cisplatin resistance was shown by Tsunoda et al. in 2005, where the authors reported a link between the receptor type 1 of IP_3_R (IP_3_R1) and increased [Ca^2+^]_i_ by cisplatin in resistant bladder cancer cell lines [37]. IP_3_R1 expression was found to be lower in cisplatin-resistant cancer bladder cells than in parental cancer cells. In addition, IP_3_R1 silencing in parental cells prevented cisplatin-induced apoptosis, while its overexpression in cisplatin-resistant cancer cells provoked apoptosis and sensitized the cells to cisplatin. The authors suggested an association of IP_3_R1 with the acquisition of cisplatin resistance in a bladder cancer model [37]. In lung cancer models, the Ca^2+^-content of the ER is lower in cisplatin-resistant cells compared to sensitive ones [18]. As for squamous lung cancer cells (EPLC), the upregulation of IP_3_R in resistant cells was observed when compared to parental ones, suggesting that IP_3_R overexpression is involved in cisplatin resistance. Interestingly, a decrease in SERCA expression was found in a small lung cancer cell line (H1339) that is resistant to cisplatin [18]. Cisplatin also induces Ca^2+^ entry from extracellular space, suggesting the implication of channels present in the PM. PM Ca^2+^ channels involve voltage-gated Ca^2+^ channels (VGCCs) but also voltage-independent Ca^2+^ channels, including ORAI and TRP (transient receptor potential) channels. VGCCs or CaVs are a family of ion channels that are activated by depolarization. There are a number of sub-types comprising the L-, N-, P-, Q-, R-, and T-types, each with varying biophysical properties [39]. This family of Ca^2+^ channels was first identified and described in excitable cells, such as neurons and muscles, but are also found in cancer cells, where they regulate oncogenic behaviors. Indeed, studies have shown that the membrane potential of cancer cells is more depolarized compared to normal cells, with a resting membrane potential varying between −30 mV and 0 mV [40]. The first evidence showing that cisplatin regulates [Ca^2+^]_i_ via an influx from extracellular compartments was published in 2007 [35] with HeLa-S3 cells. A more recent study in 2014 on MCF-7 breast cancer-derived cells showed that cisplatin modulates intracellular Ca^2+^ through the regulation of Ca^2+^ entry via VGCCs [33]. In this breast cancer cell model, the pharmacological inhibition of VGCCs by Nimodipine reduced the increase in Ca^2+^ induced by cisplatin, suggesting their involvement in the response to cisplatin. As for the study by Splettstoesser et al., the authors showed that the elevation of [Ca^2+^]_i_ in HeLa-S3 cells by cisplatin is dependent on a Ca^2+^ influx, mainly through IP_3_R-dependent PM Ca^2+^ channels. Their conclusion was based on their published data showing a reduction of Ca^2+^ influx by 2-APB, suggesting, therefore, that cisplatin may trigger the activation of IP_3_R-dependent Ca^2+^-channels located at the PM [35]. Indeed, the involvement of store-operated Ca^2+^ channels (SOC) has been more widely studied in cisplatin resistance.

Store-operated Ca^2+^ entry (Figure 2) is the main mechanism by which Ca^2+^ enters the cytosol in non-excitable cells. It is now clear that SOCE is mediated after Ca^2+^ depletion from intracellular stores through the interaction of two families of proteins: STIM (stromal interaction molecules STIM1 and STIM2) and ORAI channels (ORAI1, ORAI2, and ORAI3) [41]. SOCE is initiated when agonists bind to their receptors (G protein-coupled receptors or tyrosine kinase receptors) at the PM. The activated PLC (phospholipase C) cascade induces the hydrolysis of phosphoinositide phosphatidylinositol 4,5-bisphosphate (PIP_2_) into diacylglycerol (DAG) and inositol triphosphate (IP_3_). IP_3_ binds to its IP_3_R, situated at the endoplasmic reticulum (ER) membrane, which causes its activation and the release of Ca^2+^ in the cytosol. STIM, a single-transmembrane Ca^2+^-sensing protein, senses Ca^2+^ stock depletion and undergoes dimerization and conformational changes, allowing it to situate itself in ER-PM junctions, where it physically interacts with ORAI channels and activates Ca^2+^ entry [42].

It is now clear that SOCE dysfunction is consistently linked to cancer pathogenesis and the roles of the ORAI channels and STIM proteins have recently received more attention in cancer research [43,44,45,46]. However, their roles in response to cisplatin in different types of cancer have been poorly investigated (Table 1).

It has been demonstrated that STIM1 is upregulated in chemo-resistant osteosarcoma tissues, compared with chemo-sensitive tissues [47]. The authors showed an increase in STIM1 and SOCE in the cisplatin-resistant cell line (MG63/Cisplatin) compared to the parental MG63 line. In the parental MG63 cells, cisplatin decreased SOCE entry and increased the ER stress protein expression (GRP78, CHOP, and ATF4) leading to apoptosis, but failed to affect MG63/Cisplatin. The STIM1 knock-down, taking the siRNA approach, sensitizes resistant cells to cisplatin, while its ectopic expression in parental MG63 cells confers their resistance to cisplatin. Based on their results, the authors suggested an association of STIM1 upregulation with the cisplatin-resistant phenotype in osteosarcoma.

Two other studies reported the involvement of STIM1 in the cytotoxic effect of cisplatin in lung cancer cell models [48,49]. Li et al. have shown, using the non-small cell lung cancer (NSCLC) cell lines A549 and H460, that cisplatin induces apoptosis through SOCE by modulating STIM1 expression. In their study, they show that cisplatin decreases SOCE through a reduction in STIM1 expression. The silencing of STIM1-sensitized cells to cisplatin induced a slight SOCE decrease. Moreover, STIM1 overexpression reduced cisplatin’s apoptotic effect and had no effect on SOCE induced by cisplatin. They also reported significantly higher expression of STIM1 in lung carcinoma tissues than in adjacent non-cancerous lung tissues. They concluded that STIM1 might act as a negative regulator of apoptosis induced by cisplatin in lung cancer models. STIM1 has also been reported in the chemoresistance of NSCLC to cisplatin therapy, using the A549 cell line [49]. The silencing of STIM1 strongly decreased SOCE, along with an inhibition of the DNA response damage and a reduction in oxidative stress induced by cisplatin in A549 cells. Moreover, cisplatin had no effect on the ER Ca^2+^ content or SOCE. This result is contrary to that reported by Li et al. [48], who showed that cisplatin decreases the expression of STIM1 and SOCE. This difference between both studies could be explained by the treatment duration and the concentration of the cisplatin used.

ORAI1 expression was upregulated in several cancer models, including esophageal, squamous, ovarian, and hepatocellular carcinoma cell lines [50,51,52]. Moreover, the elevated expression of ORAI1 correlated with poor overall survival and recurrence-free survival, in esophageal squamous cancer cells, independently of other clinical parameters [50]. Using ovarian cancer cell lines, Schmidt et al. showed an enhancement of SOCE, as well as ORAI1 and STIM1 expression and Akt phosphorylation levels, in the A27080cis cisplatin-resistant cells than in the parental A2780 cell line [51]. The increased ORAI1 expression is dependent on Akt activation, since the transfection of the parental cells by a constitutive active Akt increased SOCE, while the pharmacological inhibition of Akt (SH-6) reduced SOCE amplitude only in A2780cis. Finally, late apoptosis induced by cisplatin in A2780-cis was reinforced by the pharmacological inhibitors of either Akt (SH-6) or ORAI1 (2-APB), suggesting that ORAI1 inhibition may overcome resistance. The link between ORAI1 and the PI3K/AKT pathway in chemotherapeutic agent resistance has also been reported in a hepatocellular carcinoma cell line (HepG2) exposed to 5-FU [52]. Indeed, 5-FU decreased the expression of both ORAI1 and SOCE. The overexpression of ORAI1 antagonized the effect of 5-FU, while its downregulation by siRNA decreased the SOCE and sensitized cells to 5-FU-autophagic effect by inhibiting the PI3K/AKT/mTOR pathway. One study has suggested that the downregulation of ORAI1 expression might confer resistance for the LNCaP prostate cancer cell line to cisplatin [53]. Indeed, cells lacking ORAI1 are more resistant to cisplatin’s apoptotic effects [53].

Recently, ORAI3 has emerged as an important player in malignant transformation. Indeed, ORAI3 expression was found to be upregulated in breast, lung, and pancreatic cancer tissues and regulates cell proliferation, survival, migration, and invasion [54,55,56,57]. ORAI3 is also involved in chemoresistance to cisplatin in both breast and lung cancer cell lines [58,59]. The overexpression of ORAI3 in two estrogen receptor-positive (ER^+^) breast cancer cell lines (MCF-7 and T-47D) decreased the apoptosis induced by several chemotherapeutic agents, including cisplatin, by modulating p53-protein degradation [58]. Indeed, the overexpression of ORAI3 increased the SOCE that activates the PI3K/Sgk-1 signaling pathway (Figure 3), leading to the activation of the ubiquitin ligase, Nedd4-2, probably via the Sek-1 kinase that regulates p53 degradation. Moreover, using the Gen database, ORAI3 mRNA and protein expression were higher in tumors from patients after treatment with residual disease than those from patients with partial or complete response [58]. Finally, we would like to add that the effect of cisplatin in increasing the expression of some calcium channels is not specific to ORAI3, it is also observed for ORAI1 and STIM1. Therefore, we suggest that cisplatin, by activating PI3K/Akt, increases the expression of both ORAI1 and ORAI3 in ovarian and breast resistant cancer cells, respectively.

Besides the ORAI and STIM proteins, other cation channels are involved in Ca^2+^ entry in non-excitable cells, known as TRP channels (Table 1). Transient receptor potential (TRP) channels are non-selective cation channels that are responsible for the transport of Ca^2+^, Mg^2+^, and Na^+^ under different chemical and physical stimuli. An alteration of TRP expression and/or activity has also been reported to regulate different cancer hallmarks [60,61,62]. One study using the SKOV3 ovarian cancer cell line showed the involvement of TRPC1 in cisplatin resistance. Indeed, cisplatin-resistant SKOV3 ovarian cancer cells showed a significant decrease in TRPC1 mRNA levels, compared to sensitive cells [63]. Moreover, using bioinformatic analysis, the authors showed the direct interaction of TRPC1 with PIK3C3 and SPARCL (Figure 3), two proteins that are involved in ERK-mediated autophagy. Based on this result, they suggested that TPRC1 might play a role in the development of cisplatin resistance in this cancer model. Another study on the MCF-7 breast cancer-derived cell line showed the involvement of calcium entry through TRPV1, in apoptosis, mitochondrial membrane depolarization, and reactive oxygen species (ROS) production levels induced by cisplatin. Moreover, treatment by capsazepine (a TRPV1 blocker) limited the increase of intracellular Ca^2+^ induced by cisplatin [64].


ijms-23-10687-t001_Table 1Table 1Links between Ca^2+^ homeostasis and cisplatin resistance in cancer.Ca^2+^ Channels and ProteinsCancer Types and ModelsPhenotype and Signaling PathwaysRef.
**IP_3_R1**
Bladder cancerparent cell lines (T24 and KK47), and resistant cell lines (T24/DDP10 and KK47/DDP20)Less IP_3_R1 expression in resistant cells.Silencing of IP_3_R1 in parental cells prevents apoptosis[37]
**IP_3_R**
Lung cancerResistant non-small cell lung cancer cell line (EPLC)Upregulation of IP_3_R in resistant cells, leading to a leak of Ca^2+^-content of the ER and resistance to cisplatin[18]
**SERCA**
Lung cancerResistant small lung cancer cell line (H1339)Decreased expression of SERCA in resistant cells leading to a decrease in ER Ca^2+^-content[18]
**STIM1**
Osteosarcoma(MG63/Cisplatin and MG63/WT)Increased expression of STIM1 and SOCE in MG63/Cisplatin. siSTIM1 enhanced cisplatin-induced apoptosis and ER stress in MG63/Cisplatin[47]
**STIM1**
Lung cancerA549, H460Decreased SOCE by decreasing STIM1 expression leading to apoptosis. Potential involvement of STIM1 in the microtubule cytoskeleton during mitosis[48]
**STIM1**
Lung cancer (A549)Cervical cancer (HeLA)Expression of STIM1 is higher in carcinoma tissue than in the adjacent non-tumor lung tissueSOCE contributes to ROS production, DNA damage response, and apoptosis in response to cisplatin. SOCE inhibition reduces cisplatin-induced ROS production and apoptosis[49]
**ORAI1**
Prostate cancer (LNCaP)Cisplatin increased ORAI1 expression, leading to an increased basal Ca^2+^ concentration likely through ORAI1.ORAI1 KD decreased the apoptosis rate induced by cisplatin[53]
**ORAI1 and STIM1**
Ovarian cancerA2780cis/A2780Increased expression of ORAI1 and STIM1, leading to increased SOCE. Increased Akt activation leads to the upregulation of ORAI1, contributing to cisplatin resistance[51]
**ORAI3**
Breast cancerMCF-7, T-47DORAI3 mRNA level was higher in tumor tissues, showing residual diseaseCalcium entry through ORAI3, activation of PI3K, Sgk-1, Sek-1, and Mdm2, leading to p53-protein degradation and apoptosis resistance[58]
**TRPV1**
Breast cancerMCF-7TRPV1 activation, with increased Ca^2+^ concentration leading to apoptosis and oxidant effect[64]
**TRPC1**
Ovarian cancerSKOV3-CisTRPC1 is less expressed in resistant SKOV3-cis compared to sensitive SKOV3 cellsTRPC1 forms a complex with PIK3C3 and SPARCL mediating autophagy through ERK[63]


## 3. Cisplatin-Based Chemotherapy and the Emergence of CSC Markers

Cisplatin-based chemotherapy plays an important role in the treatment of cancer. Unfortunately, the resistant phenotype that occurs after cisplatin treatment is a major therapeutic challenge. One way to explain this resistance is based on the emergence or expansion of resistant cells carrying stem cell markers called CSCs. Before we explain how cisplatin participates in CSC marker emergence, we will briefly define what CSCs are and present hypotheses about their origin at tumor initiation.

### 3.1. CSC Definition and Origin at Tumor Sites

Stem cells are characterized by the ability to self-renew and are capable of differentiating into particular and specialized cell types. Through self-renewal, more stem cells are produced, thereby maintaining an undifferentiated state. By differentiation, stem cells give rise to a mature specialized cell type. While embryonic stem cells (ESCs) are capable of differentiating into all tissues during embryonic development, adult stem cells play an important role in repairing and replacing adult tissues [8,65,66]. Research studies showed that a subpopulation of stem-like cells within tumors shares the same characteristics as stem cells, known as CSCs. CSCs are a subpopulation of cells inside the cancer microenvironment niche that define a reservoir of independent cells with the unique capability of self-renewal, multi-potent tumor-initiating properties, and of assembling various lineages of cancer cells [66]. Although CSCs exhibit similar characteristics to normal stem cells, the hypotheses regarding their origin are still unclear. It has been suggested that the origin of CSCs at tumor initiation is driven by transformed differentiated cells or by transformed tissue-resident stem cells. The proliferation and differentiation of adult tissue-resident stem cells are a part of the physiological regeneration program that maintains tissue homeostasis. Adult tissue-resident stem cells can divide asymmetrically and generate transient proliferating cells, which possess a high amplifying capacity [16,17]. These cells are said to terminally differentiate, a process in which they will eventually lose their proliferative capability, thereby maintaining organ homeostasis. In cancers, tumors can arise by accumulating several mutations that transform differentiated cells and cause a de-differentiation; therefore, progenitor cells or terminally differentiated cells could revert to a stem-like phenotype following oncogenic mutation, reactivating self-renewal genes that were previously lost upon differentiation [67]. On the other hand, tissue-resident stem cells, as well as their progeny, can endure genomic changes and accumulate mutations that lead to uncontrolled growth and niche-independent development. Altogether, this results in the generation of heterogeneous tumors [66].

### 3.2. CSC Markers in Different Types of Cancers

CSCs were found to share similar markers as the ones found in adult stem cells such as ALDH (aldehyde dehydrogenase), and the transcription factors Nanog, Oct-4, and SOX-2. The three latter factors are important for maintaining the self-renewal property and ALDH is known as the main detoxification enzyme [9,68,69]. Research studies have proven that CSCs from the breast [70] and pancreas [71] are commonly CD44^+^ CD24^low^, while CSCs from the brain [72], colon [73], prostate [74], pancreas [75], and lung [76] have been found to be CD133^+^. However, it is important to mention that CSCs markers are sometimes debatable; as an example, although glioblastoma CD133^+^ cells are highly tumorigenic, some CD133^−^ cells were still able to form a tumor when grafted into mice [77]. In addition, CD133 expression seems to be a poor prognostic marker in patients [78]. In fact, the attribution of specific markers to stem cells of a particular tumor is performed after a series of experiments proving its validity. Several groups used the established markers to sort the cells from freshly dissociated tumor tissues, culturing them in specific conditions and transplanting them into mice to evaluate their ability to form a tumor. Different surface markers that are frequently used to isolate CSCs cell subpopulations were reviewed by Phi and colleagues [79] and by Visvader and Lindeman [80].

### 3.3. Chronic Cisplatin Exposure and Stem Cell Marker Induction

The hypothesis to explain how cancer cells can resist chemotherapeutic agents is based on two mechanisms called “innate chemoresistance “or “acquired chemoresistance”. In “innate chemoresistance”, the cancer cells possess some mechanisms responsible for the elimination of the cytotoxic effect of chemotherapeutic drugs (for review, see [81]) while in “acquired chemoresistance”, chronic exposure to the drug treatment is able to enrich the cancer cell population with a subpopulation possessing CSC specifications, such as the increased expression and/or activity of the cellular detoxification proteins.

Liu and colleagues [80] showed that cisplatin treatment for 24 h remarkably increased the percentage of CD133^+^ subpopulation cells in two non-small cell lung cancer (NSCLC) cell lines (H460 and H661). They demonstrated that CD133^+^ enrichment was specifically achieved by cisplatin or carboplatin exposure, but not by doxorubicin and paclitaxel. Functional assays showed that CD133^+^ cells displayed greater sphere-forming ability, cisplatin resistance, and migration capacity. The fact that the pharmacological blockade of the Notch-1 pathway by DAPT prevented the enrichment of CD133^+^ cells induced by cisplatin, suggested that this pathway might constitute a key component in the CSC phenotype appearance induced by chronic exposure to cisplatin in lung cancer. In addition to cisplatin, carboplatin is also used as a platinum-based chemotherapy agent for NSCLCs, especially for patients with chronic renal diseases. The authors found that carboplatin treatment also triggered DNA damage and induced the enrichment of CD133^+^ cells that express higher mRNA levels of stem cell-associated genes, including Oct-4, SOX-2, Nanog, Smo, Bmi1, β-catenin, Nestin, and Notch1, along with the multidrug-resistant genes ABCG2 and ABCB1 [80]. MacDonagh and colleagues [82] also found that chronic cisplatin exposure is able to produce the expansion of an ALDH1 activity-positive subpopulation in NSCLC cell lines. The cisplatin-resistant cells showed greater ALDH1 activity than the parental cell lines. Moreover, the ALDH1^+^ sub-population that was sorted from cisplatin-resistant cells showed an increase in Nanog and Oct-4 gene expression, and CD133 was upregulated. Interestingly, ALDH1 inhibition reversed the resistance and re-sensitized the NSCL cells to cisplatin.

## 4. The Role of Ca^2+^ in Cancer Stemness and Resistance to Cisplatin through CSC Population Enrichment and via the Ca^2+^-Dependent Signaling Pathways

### 4.1. The Role of Ca^2+^ and Ca^2+^ Channels in Cancer Stemness

Recent studies have found that changes in intracellular Ca^2+^ signaling are important in the self-renewal process and stem cell behaviors (proliferation and differentiation) [16,17,83,84], with the involvement of a wide variety of Ca^2+^ channels. Therefore, aberrant Ca^2+^ signals could be responsible for CSCs-mediated chemoresistance and may represent an efficient targeting approach. CSC development is controlled by intracellular factors and extracellular molecules from the microenvironment, which stimulate various sets of receptors. Some of these extracellular molecules participate in inducing either local transient elevations of intracellular Ca^2+^ or propagated prolonged Ca^2+^ signals throughout the whole cell. Consequently, cell behaviors depend on the nature and magnitude of those Ca^2+^ signals that are dictated in a spatiotemporal manner [15,85]. Increases in free cytosolic Ca^2+^ concentrations do not only occur from an extracellular Ca^2+^ influx but it also requires a Ca^2+^ release from intracellular stores mainly reserved in the ER and mitochondria. Since Ca^2+^ alterations are observed as a result of the abnormal expression and/or activity of Ca^2+^ channels, similar to those observed in oncogenesis [11,13,43,45,46,86,87]; it is suggested that in CSCs, changes in Ca^2+^ signals due to aberrant channel expression could be advantageous for CSC oncogenic behaviors, thus promoting cancer development and resistance. Consequently, the implication of channels in this process represents a crucial tool to target CSCs. In this section, we explore this hypothesis by overviewing a global outline of the Ca^2+^ channels linked to the CSC function (Table 2).

In CSCs, higher depolarized resting membrane potential has been observed, suggesting that VGCCs may be involved in cancer stemness [88]. For example, in hepatocellular carcinoma, CSCs display more depolarized resting membrane potential compared to normal stem cells. However, to date, CaV3.2 is the only T-type VGCC (Table 2) described in glioblastoma CSCs, where its upregulation was observed and associated with proliferation [89]. The pharmacological modulation of this channel using mibefradil inhibited glioblastoma stem cell proliferation and induced apoptosis [90]. CaV3.2 was also described as favoring the differentiation of glioblastoma CSCs. When CaV3.2 expression was reduced, stem-related genes such as Nestin, CD133, and SOX-2 were downregulated, while the astrocyte differentiation marker GFAP (glial fibrillary acidic protein) was upregulated. The authors showed that when the CaV blocker mibefradil was combined with temozolomide, an efficient chemotherapy agent that is currently used to treat glioblastoma, tumor growth was reduced in mouse xenograft models and was associated with a prolonged survival rate. While CSC resistance is a major obstacle in current oncology practice, these studies suggest that targeting VGCCs could further improve current therapies and prevent the resistant CSC subpopulation from relapsing after successful initial treatment. In normal mouse embryonic stem cells, CaV3.2 knockdown or blockage with nickel cations (Ni^2+^) induced a loss in self-renewal capacities and reduced the expression of stem cell genes Oct-3, Oct-4, and Nanog, indicating that the CaV3.2 function in stem cell functions is conserved [91].

As for PM SOCE channels (Table 2), a role for ORAI1 in oral/oropharyngeal squamous cell carcinoma (OSCC) stem cells has been established [92]. This study showed that ORAI1 expression increased with tumor progression and that it was also upregulated in CSC populations of OSCC. In ALDH^+^ CSC subpopulations, ORAI1 was elevated and promoted stemness through NFAT. When these cancer cells were transfected with a dominant negative mutant of ORAI1, sphere formation and tumor development in xenografts were considerably inhibited. On the other hand, the ectopic expression of ORAI1 in non-tumorigenic immortalized oral epithelial cells increased proliferation, self-renewal, and tumor-promoting capacities, implying the functional importance of Orai1/NFAT axis in OSCC CSC regulation. Furthermore, gene analysis performed to decode the implications of Ca^2+^ signaling in brain tumor samples showed that ORAI1 is highly expressed in glioblastoma tissues and in glioma stem cells but not in normal brain tissues [93]. This suggests that ORAI1-mediated SOCE may play a crucial role in maintaining or enriching stem cell populations. More recently, research studies showed that in glioblastoma, changes in Ca^2+^ homeostasis during the switch from proliferation to quiescence state are controlled by SOC channels since the inhibition of SOC channels with the pharmacological agent SKF-96365 drives proliferating GSLC cells to quiescence. This switch is characterized by an increased capacity of GSLCs mitochondria to change morphology and capture Ca^2+^. These data suggest that the remodeling of Ca^2+^ homeostasis and the reshaping of mitochondria might favor the survival of quiescent GSLCs [94]. The role of STIM1 has not yet been fully understood in CSCs, but it is known that under the hypoxic conditions of the tumor microenvironment, stemness and resistance to anticancer therapies are promoted in CSCs [17]. In hepatocellular carcinoma cells, for instance, the hypoxia-inducible factor-1 alpha (HIF-1α) directly controls STIM1 transcription and contributes to SOCE. In parallel, Ca^2+^ influx, induced by increased SOCE, increased HIF-1 accumulation in hypoxic cells, suggesting a direct link between SOCE and hypoxia. If so, it is not surprising that these proteins could promote stem cell-like phenotypes [95].

As for TRP channels (Table 2), at this point, only a few members of the TRP family have been linked to stem cell function; these are TRPC3, TRPA1, TRPV1, TRPV2, and TRPM7. Most recently, Hirata et al. demonstrated that lysosphingolipid sphingosine-1-phosphate mediates the CSC phenotype, which can be identified as the ALDH-positive cell population in several types of human cancer cell lines. In addition, they found that lysophosphatidic acid (LPA) receptor 3 was highly expressed in ALDH^+^ triple-negative breast cancer cells. Mechanistically speaking, the LPA-induced increase in ALDH^+^ cells was dependent on an increase in cytosolic Ca^2+^, which was suppressed by a selective inhibitor of TRPC3. Moreover, IL-8 production was involved in the LPA response via the activation of the Ca^2+^-dependent transcriptional factor nuclear factor of activated T cells [96].

TRPV2 has been directly associated with the self-renewal pathway of CSCs. A microarray analysis of CSCs isolated from esophageal squamous cell carcinoma tumors showed that these CSCs showed the significantly upregulated expression of TRPV2; this observation was confirmed by PCR when compared to control cells [97]. When TRPV2 was inhibited by the pharmacological agent, Tranilast, CSC populations were significantly reduced, and their self-renewal capacity was abolished. In contrast, in other cancer models, a reverse correlation between TRPV2 expression and the tumorigenic aspect of CSCs was observed, where the downregulation of TRPV2 enhanced stem cell function [98]. For example, TRPV2 expression was decreased in liver CSC populations compared to normal tissues; this downregulation was associated with an increase in stem cell markers such as CD44, CD133, and ALDH1. Knockdown of TRPV2 promoted stem cell enrichment while its overexpression reversed this phenomenon and reduced stem cell markers. This course was also observed in one liver cancer in vivo model, where the reduction of TRPV2 expression enhanced tumor development, while its overexpression displayed the opposite effect [99]. In agreement with the previous observations, another study demonstrated that TRPV2 expression was downregulated in glioblastoma stem cells compared to differentiated cells [100]. Higher TRPV2 expression in differentiated cells leads to an increase in basal Ca^2+^ levels, as well as the expression of a differentiation marker, and reduces the stem pathway, Nestin. This was confirmed using both pharmacological and siRNA approaches, showing a decrease in the expression of the differentiation marker, GFAP, whereas the overexpression of TRPV2 reversed the observed effects. Differentiated cells were then treated with cannabidiol (CBD), which is known to activate TRPV2, leading to an increase in basal Ca^2+^. After using CBD to stimulate TRPV2 expression and activity in glioblastoma stem cells, a reduction in stem cell proliferation and self-renewal properties was detected, correlating with an increase in the terminal differentiation marker, GFAP, and a decrease in the stem cell markers CD133 and Oct-4. When CBD treatment was combined with the conventional chemotherapy agent, Carmustin, the apoptosis of glioblastoma stem cells was enhanced and improved outcomes regarding tumor progression were obtained. Similar findings were also observed in xenografted mouse models when TRPV2 overexpression reduced tumor volume and promoted the differentiation of stem cells [100]. The same observations have also been described in glioblastoma cells, where the overexpression of TRPA1 and TRPV1 caused cell differentiation, inducing apoptotic cell death and the loss of stemness [101]. Taken together, these data suggest that TRPV1, TRPV2, and TRPA1 could be considered potential CSC therapeutic targets to impair stemness. As for TRPM7, two research works have also demonstrated TRPM7 expression in both glioblastoma and neuroblastoma stem cells, whereas TRPM7 has been linked to stemness promotion and metastasis. In glioblastoma cells, high levels of TRPM7 expression were observed compared to the differentiated controls, and its downregulation significantly reduced the stem cell markers CD133 and ALDH1 and inhibited two key stem pathways, STAT3 and Notch [102]. In neuroblastoma cells, the upregulation of TRPM7 was associated with increased metastatic behavior. The downregulation of TRPM7 expression in the SH-SY5Y neuroblastoma cell line, as well as microarray and studies of bioinformatics, were performed [103]. The results suggested that TRPM7 induced development and differentiation by activating the genes and pathways that are directly linked to stemness, such as STAT3, Wnt1, Notch1, and SNAI1. Increased TRPM7 expression was associated with enhanced SOX2, KLF4, CD133, Hsp90α, uPA, and MMP2 expression in lung cancer stem cells. TRPM7 silencing inhibited the epithelial-to-mesenchymal transition (EMT), suppressed stemness markers and phenotypes, and simultaneously inhibited the Hsp90α/uPA/MMP2 axis. When TRPM7 was treated with the inhibitor Waixenicin A, lung cancer cells failed to form tumor spheres in vitro [104].

As for intracellular Ca^2+^ reservoirs and transporters (Table 2), their function in controlling CSC stemness has been investigated in few studies. Among the activated pathways leading to intracellular Ca^2+^ signaling is the release from the endoplasmic reticulum (ER) to the cytoplasm through ryanodine receptors (RYR) and/or the inositol 1,4,5-trisphosphate receptors (IP_3_R). IP_3_Rs and RYRs constitute four subunits with a regulatory cytoplasmic domain. There are three isoforms of IP3Rs—IP3R1, IP3R2, and IP3R3—as well as RYRs (RYR1, RYR2, and RYR3). Their expression and functions have not been widely described in cancer stem biology until recently. In breast cancer stem cells, RYR1 promoted Ca^2+^ release after treatment with a carboplatin agent. In fact, HIF-1-dependent glutathione S-transferase omega 1 (GSTO1) expression was induced, thereby interacting with the ryanodine receptor RYR1, leading to its activation. Consequently, increased cytosolic Ca^2+^ levels enhanced Nanog expression and activated the intracellular signaling cascade, PYK2 (protein tyrosine kinase 2b)/SRC /STAT3 (signal transducer and activator of transcription 3), and enhanced CSC population enrichment [105]. In addition, RYR1 knockdown decreased the percentage of ALDH^+^ CSCs and reduced carboplatin-induced pluripotency factor expression, as well as tumor initiation and volume after injection in mice. Another study demonstrated that RYR downregulation promotes a stem-like phenotype in medulloblastoma, whereas microRNA miR-367 expression was upregulated, thereby decreasing RYR3 expression and enhancing spheroid culture formation. This phenomenon showed an increase in self-renewal capacities, along with the activation of stem pathways and markers, such as Oct-4 and CD133, respectively [106]. Ca^2+^ release through IP_3_R was shown to play an important role in melanoma CSCs, when the reduction of IP_3_R maturation after selenoprotein k (SELENOK) knockdown reduced Ca^2+^ release and led to a decrease in CD133^+^ populations [107]. In vivo, these observations were confirmed using a transgenic mouse model that developed spontaneous metastatic melanoma after crossing with SELENOK CRISPR/Cas9 knockout mice. The SELENOK-deficient mice, which were born from the same litter, displayed reduced primary tumor growth, suggesting that Ca^2+^ release through IP_3_R is required for melanoma stemness and tumor development. Ca^2+^ release from IP_3_Rs also seems to be involved in quiescence, a phenotype allowing CSCs to evade and escape from anti-cancer therapies.
ijms-23-10687-t002_Table 2Table 2Roles of Ca^2+^ and Ca^2+^ channels in cancer stemness.Ca^2+^ ChannelsCancer TypeFunction in Cancer Stem Cell BiologyReferences**T-type:****CaV3.2**GlioblastomaIncrease in CD133^+^ stem population. CaV3.2 reduces stem cell properties and sensitizes CSCs to chemotherapy treatment[90]**ORAI1**Oral/oropharyngeal squamous cell carcinoma (OSCC)ORAI1 expression is high in CSCs and linked to self-renewal, migration, and transcription factor expression, such as Nanog and OCT3/4, through activation of the NFAT pathway[92]**TRPA1**GlioblastomaOverexpression leads to cell differentiation, inducing apoptotic cell death and the loss of stemness[101]**TRPC3**Breast CancerOverexpression of TRPC3 enhances cell renewal abilities through the increase of IL-8 secretion via the LPA/LPAR3/TRPC3 pathway[96]**TRPV1**GlioblastomaOverexpression leads to cell differentiation, inducing apoptotic cell death and the loss of stemness[101]**TRPV2**EsophagealLiverGlioblastomaALDH^+^ stem cells have high TRPV2 expression. Inhibition reduces stem markers and self-renewalLiver cancer stem cells have low TRPV2 expression. Overexpression reduces stem cell markers and self-renewal capacityOverexpression of TRPV2 induces CSC differentiation and reduces self-renewal capacity. CBD-mediated activation of TRPV2 enhances chemotherapy and sensitizes the death of CSCs[97,98,99]**TRPM7**GlioblastomaNeuroblastomaLung CancerTRPM7 expression is high in stem cell populations, along with STAT3 and survivin. TRPM7 is required for stem cell self-renewal and differentiation.TRPM7 is associated with development and differentiation pathways, promoting the expression of genes such as STAT3, Wnt1, and NOTCH1.Overexpression leads to tumor sphere formation and stemness maintenance through the activation of the Hsp90α/uPA/MMP2 signaling pathway[102,103,104]**RYR1**Breast cancerRYR1 silencing reduces stem cell populations and Nanog[105]**RYR3**MedulloblastomaDownregulation of RYR3 enhances Nanog and CD133 expression and promotes spheroid formation[106]**IP_3_R**MelanomaA decrease in IP_3_R expression probably mediates a decrease in calcium flux as well as stem pathways[107]


### 4.2. Role of Ca^2+^ Channels in Cisplatin Resistance through the Increase of Cancer Stem Cell Populations

Several studies described the involvement of cancer stem cells in resistance to chemotherapy but only two of them reported a direct link between Ca^2+^ channels, resistance to cisplatin, and cancer stem cells (Table 3). The first study described the link between cells expressing the VGCC α2δ1 subunits (α2δ1^+^) and resistance to cisplatin in small cell lung cancer (SCLC) models [108]. VGCCs are composed of pore-forming α1 and auxiliary β and α2δ subunits [109]. The α2δ subunits are important auxiliary subunits of several types of VGCCs [110] and can increase the VGCC density on the plasma membrane and regulate Ca^2+^ currents [111,112]. The SCLC type is less common than NSCLC but is considered to be the most aggressive form of lung cancer. α2δ1-positive cells (α2δ^+^), isolated from three SCLC cell lines and patient-derived xenograft models, showed greater sphere formation ability, growth rate, and tumorigenicity. In addition, this population showed higher stemness transcription factors (CD133, SOX-2, Oct-4, and Nanog) and multi-drug-resistant genes (MDR1 and ABCG2) than parental cells (α2δ1^−^ cells). Treatment with a chemotherapy regimen comprising cisplatin and etoposide induced an increase in the α2δ1^+^ population and increased chemoresistance. Furthermore, pERK1/2 was upregulated, and SOX-2 was overexpressed in α2δ1^+^ H1048 cells. The ectopic overexpression of α2δ1 in the H1048 cell line induced high levels of p-ERK, CSCs-related genes (BMI1, NANOG, SOX2, and BCL2), and drug resistance genes (ABCG2 and MDR1), and a decrease in proapoptotic protein BAX and cleaved caspase-3. Finally, the inhibition of the ERK pathway by a pharmacological compound or by using an α2δ-neutralizing antibody sphere formation was decreased in α2δ1^+^ cells. Interestingly, the combined ERK inhibitor and the α2δ-neutralizing antibody treatment strongly inhibited the growth of α2δ1^+^ cells [108]. They also exposed several PDX models with different CD133 and α2δ1expression levels to chemotherapy. Despite the fact that PDXs with different stem cell marker expressions showed diverse responses to chemotherapy, they found that the models with low α2δ1 expression or a low α2δ1/CD133 ratio showed a good efficiency in chemotherapy, contrary to those of high α2δ1 expression and α2δ1/CD133 ratio, which were unresponsive to the same treatment. The exposition of resistant-PDXs to chemotherapy combined with α2δ1 monoclonal antibody showed an improved response compared to the group treated with chemotherapy alone. It should be noted that the authors did not investigate the involvement of Ca^2+^ influx in the resistance induced by cisplatin. Another study using two CSC-enriched ovarian cancer cell lines (A2780-SP and SKOV3-SP) [113] showed an upregulation of CSC markers that are associated with higher resistance to both paclitaxel and cisplatin (Table 3). The pharmacological inhibition of the voltage-gated Ca^2+^ channel (VGCC) by manidipine, lacidipine, benidipine, or lomerizine reduces stemness and CSCs apoptosis, and inhibits the AKT and ERK signaling pathways. In these CSC cells, the expression of 3 genes (namely, CACNA1D, CACNA1F, and CACNA1H), coding for the L/T Ca^2+^ channels, is more upregulated in ovarian CSCs than in ovarian cancer cells. Silencing these three VGCC genes reduces stemness marker expression at a similar level obtained with the pharmacological inhibitors, as well as ERK phosphorylation. Moreover, the increased expression of L- and T-type Ca^2+^ channel subunits in ovarian cancer tissue was correlated with a poor prognosis. Furthermore, when cisplatin treatment was applied alone, it did not suppress the A2780-SP cells’ viability and proliferation, but its combination with any of the four Ca^2+^ channel blockers suppressed both CSCs viability and proliferation more strongly. Taken together, they suggested that these VGCC inhibitors might be used as potential therapeutic drugs for preventing ovarian cancer recurrence.

As for ORAI channels, very recently, Ouadid-Ahidouch’s group demonstrated clearly the involvement of ORAI3 in the enrichment of the CSC population by cisplatin treatment in lung cancer cell models, which, in turn, induces resistance to cisplatin [59] (Table 3). In their study, ORAI3 expression was found to be higher in lung adenocarcinoma tissues when compared to adjacent normal tissue and was correlated with higher tumor grades. Furthermore, ORAI3 was suggested as a prognostic marker of metastasis and survival in lung adenocarcinoma cell models. ORAI3 was shown to regulate cell proliferation in two NSCLC cell lines through the AKT pathway. Besides, using an immunohistochemical assay, Daya et al. showed that there is greater ORAI3 staining in lung adenocarcinoma tissue biopsies taken from patients after chemotherapy, compared to the staining observed in the tissues taken before chemotherapy [59]. Moreover, patients whose biopsies showed higher ORAI3 expression levels revealed partial or even no response to the given chemotherapy. In two adenocarcinoma cell lines (A549, H23), they found that cisplatin treatment induced the upregulation of ORAI3 and an increase in SOCE. Cells overexpressing ORAI3 possessed resistance to cisplatin-induced apoptosis; those downregulating ORAI3 were prone to cellular apoptosis, particularly due to cisplatin exposure. Cisplatin-induced apoptosis resistance was accompanied by an enhancement of the CSC population via the increased expression of CSC markers (Nanog and SOX-2). Extracellular Ca^2+^, as well as ORAI3, were able to modulate the expression of these markers. Moreover, the PI3K/Akt pathway was involved in increased ORAI3, as well as CSC marker expression by cisplatin.

### 4.3. Ca^2+^-Dependent Signaling Pathways Involved in Cisplatin Resistance in CSCs

A major cause of chemoresistance is cancer stemness. For example, the chemoresistance of primary and acquired CSCs often involves an enhanced drug efflux. This is achieved by increased ATP-binding cassette (ABC) membrane transporters (such as MDR1), ABC sub-family G member 2/3 (ABCG2/3), and ABC sub-family C member 2/4/5 (ABCC2/4/5), which are downstream of the cancer stemness-related genes SNAIL, SLUG and TWIST [114,115,116,117,118]. Those transcription factors are able to induce epithelial cells into the mesenchymal phenotype via EMT and produce chemoresistance by extension [119]. One oncogene pathway [120] having the properties to activate those transcription factors is STAT3 [121,122,123,124]. In this case, trying to block STAT3 could be a promising strategy to avoid chemoresistance. It is well known that STAT3 signaling could be activated via the canonical transduction signaling pathway, by means of JAK2 and SRC activation. However, recently, Shiratory-Hayashi and colleagues demonstrated that IP_3_R1/TRPC channel-mediated Ca^2+^ signals that are elicited by IL-6 are necessary for persistent STAT3 activation in astrocytes [125]. Up until now, this finding was not described for stem cells, but it could suggest a hypothesis regarding the role of Ca^2+^ influx in STAT3 activation and the acquisition of a chemoresistance phenotype.

The p53 protein is a well-known key actor to permit cell cycle entrance. In case of DNA damage, p53 contributes to cell cycle arrest and could be considered in differentiated cells as a tumor suppressor actor. Functionally, p53 is a transcription factor and may play an important role in stem cell phenotype appearance [126]. Subsequent publications further confirm the finding that p53 is highly abundant in embryonic stem cells [127,128]. In addition to controlling cell proliferation, p53 regulates epithelial-mesenchymal transition. The link between p53 and cell differentiation was evoked by Lin et al. [129]. They found that p53 is able to bind to the promoter of Nanog and suppresses its expression. Nanog expression is required for self-renewal and the maintenance of undifferentiated states [130,131]. In cancer, the first study highlighting the link between Ca^2+^ influx, p53 activity, Nanog expression, and chemoresistance was performed by Hasna and colleagues in 2018 [58]. The authors demonstrated that the Ca^2+^ influx mediated by ORAI3 is able to regulate p53 activity and allow resistance to chemotherapeutic drugs in breast cancer cells (Figure 3). Daya et al. [59] suggested that this Ca^2+^ influx, mediated by ORAI3, could participate in increased Nanog expression and stem-cell-like phenotype appearance in lung cancer models (Figure 3). Thus, ORAI3 overexpression studies could be a key step in understanding how this Ca^2+^ channel or Ca^2+^ influx could activate chemoresistance processes.

## 5. Conclusions

The use of cisplatin in cancer therapy has significantly reduced morbidity and improved survival over the past few decades. Despite this progress, its recurrence remains high and presents a clinically relevant pattern of failure in many malignancies. Despite the effectiveness of cisplatin in oncology, resistance to this agent constantly hampers its therapeutic success. The mechanisms underlying this phenomenon remained poorly understood until very recently, when stemness emerged as one of the mechanisms involved in the resistance process. Ca^2+^ channels are now considered to be key players in CSC biology and chemoresistance, but the studies on their functions in this process are still burgeoning. Based on the published data mentioned in this review, demonstrating that the modulation of Ca^2+^ channels can overcome resistance by sensitizing cancer cells to cisplatin-mediated cell death, it is promising to adopt novel therapeutic strategies. Gene-targeted or mRNA-based therapies could be useful tools to impair CSC functions as they are now used as an alternative option to block Ca^2+^ channels and treat Ca^2+^ channel-related cardiovascular diseases, for example [132,133]. However, these methods must be validated using a pharmacological approach; therefore, the development of nanocarriers to improve both ion channel modulation and cisplatin bioavailability should be considered. The possibility of targeting SOCE has already been reported in several reviews [134,135]; but since Ca^2+^ channels are expressed in the majority of cancers, it is difficult to distinguish whether SOCE targeting would impact cancer cells or CSCs. One therapeutic option would be to target the channels that are either significantly downregulated or strongly upregulated in CSCs and that regulate specific signaling pathways. For instance, inhibiting voltage-gated Ca^2+^ channels (L/T-type) with manidipine could overcome chemoresistance by suppressing stemness, as shown in ovarian cancer models [113]. In addition, mibefradil, a T-type Ca^2+^ channel blocker that was initially developed as a cardiovascular drug, was recently found to strongly inhibit the ORAI3 channels in HEK-293T-Rex cells [136]. Thus, the combinational use of cisplatin with mibefradil could overcome ORAI3-relative resistance, as demonstrated in breast or lung cancer models. Another SOCE inhibitor, RP4010, a compound that was developed by Rhizen Pharmaceuticals SA, is in phase I/Ib clinical trials to evaluate its safety and efficiency in patients with relapsed or refractory lymphomas (NCT03119467). RP4010 has been also shown to reduce SOCE by inhibiting ORAI1 and the PI3K/Akt pathway when combined with chemo agents (gemcitabine and nab-paclitaxel) in pancreatic ductal adenocarcinoma cells [137]. It would be also interesting to synthesize and develop pharmacological compounds that target TRPV1, TRPV2, and TRPA1 that were shown to be downregulated in some types of cancers and whose activity enhances differentiation and impairs stemness. Finally, although the PI3K pathway seems to be a common track involved in increasing the stemness marker expression of cisplatin-exposed cancer cells, it seems clear that this is not the case for all Ca^2+^ channels affected by cisplatin resistance among cancer types. Thus, further investigations are required to characterize a Ca^2+^ signature for CSCs functions, as well as their associated signaling pathways, in order to prevent cancer reappearance.

## Figures and Tables

**Figure 1 ijms-23-10687-f001:**
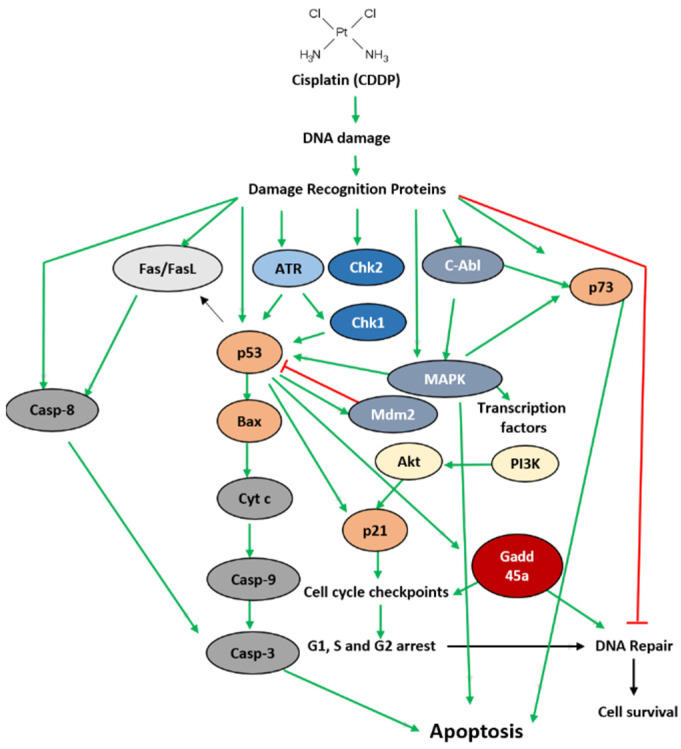
A brief overview of the pathways involved in mediating cisplatin-induced effects in cells. Apoptosis as well as cell survival depend on the nature and intensity of the signals generated and the crosslinks between the pathways involved. Apoptosis is initiated following the recognition of DNA damage by candidate proteins that bind to physical distortions in the DNA induced by cisplatin. When DNA damage is detected and cannot be repaired, the activation of the irreversible intrinsic death pathway mediated by p53 occurs. Depending on the intensity of DNA damage, the cell cycle is arrested at checkpoints that prevent its progression for DNA repair and cell survival. Abbreviations: ATR: ataxia telangiectasia and Rad3-related protein; Akt: protein kinase B; Bax: Bcl-2–associated X; Chk1/2: checkpoint kinases 1 and 2; C-Abl: tyrosine-protein kinase ABL1; casp: caspase; Cyt c: cytochrome c; Fas/FasL: Fas–Fas Ligand; GADD45a: growth arrest and DNA damage-inducible alpha; MAPK: mitogen-activated protein kinases Mdm2: mouse double-minute 2; PI3K: phosphoinositide 3-kinase; p73: tumor suppressor p53-related protein; p53: tumor suppressor protein; p21: cyclin-dependent kinase inhibitor.

**Figure 2 ijms-23-10687-f002:**
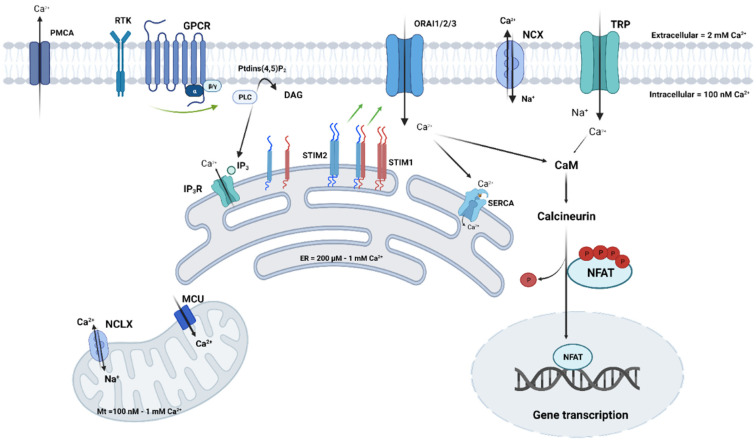
Ca^2+^ homeostasis in non-excitable cells. Agonists bind to their receptors (GPCR or TK), which are located at the plasma membrane (PM), thereby activating PLC. The hydrolysis of phosphatidylinositol 4,5-biphosphate leads to the production of IP_3_ and the release of Ca^2+^ from the ER store via IP_3_R. ER store depletion activates the STIM proteins, which translocate to ER–PM junctions and gate heterohexamer ORAI channels. Ca^2+^ influx activates a set of proteins and effectors that are responsible for gene transcription. Calcium homeostasis is maintained by PM and organelle channels, transporters, and exchangers. Abbreviations: CaM: calmodulin; DAG: diacylglycerol; ER: endoplasmic reticulum; GPCR: G protein-coupled receptor; IP3R: inositol 1,4,5-trisphosphate receptor; Mt: mitochondria; MCU: mitochondrial Ca^2+^ uniporter; NCX: Na^+^/Ca^2+^ exchanger; NCLX: Na^+^/Ca^2+^/Li^+^ exchanger; NFAT: nuclear factor of activated T-cells; PLC: phospholipase C; PMCA: plasma membrane Ca^2+^ ATPase; PtdIns(4,5)P2: phosphatidylinositol 4,5-bisphosphate; RTK: receptor tyrosine kinase; SERCA: sarco-/endoplasmic reticulum Ca^2+^-ATPase; STIM: stromal interaction molecule. This figure was created using www.biorender.com.

**Figure 3 ijms-23-10687-f003:**
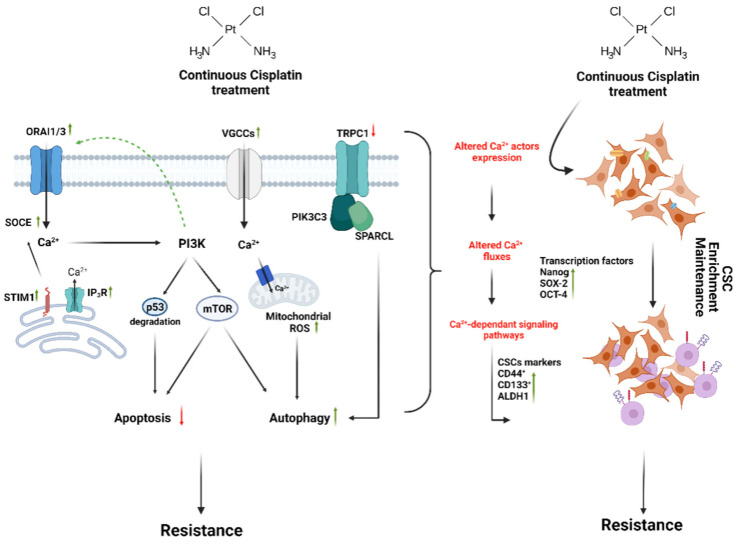
Summary of cisplatin-mediated resistance through Ca^2+^ signaling and stemness. Chronic cisplatin exposure affects the expression of Ca^2+^ channels and other Ca^2+^ homeostatic proteins. On the one hand, the altered Ca^2+^ fluxes activate Ca^2+^-dependent pathways that are involved in the survival of cancer cells, such as the inhibition of apoptosis and enhancement of autophagy. On the other hand, stem-related Ca^2+^-dependent transcription factors are activated, participating in the enrichment of CSC populations and their maintenance, which promotes tumor re-development. This figure was created with www.biorender.com.

**Table 3 ijms-23-10687-t003:** Ca^2+^ signaling is responsible for cisplatin resistance through stemness.

Ca^2+^ Channels	Cancer Type and Models of Study	Function in Cancer STEM Cell Biology	Reference
**α2δ1** **(VGCC)**	Lung Cancer NCI-H1048, NCI-H69, and NCI-H209 Several PDX models	Chemotherapy increased the expression of α2δ1^+^ CSCs population, pERK, CSCS, and drug resistance markers and downregulated BAX and cleaved caspase-3. Inhibition of pERK or/and α2δ1 subunits reduced sphere formation. PDX showing a high expression of α2δ1 was not sensitive to cisplatin/etoposide treatment in contrast to PDX, showing less α2δ1 expression	[108]
**Voltage-gated Ca^2+^ channels (T/L types)**	Ovarian cancer (A2780-SP and SKOV3-SP)	L- and T-type Ca^2+^ channel subunit expression is upregulated in ovarian cancer tissue and is correlated with poor prognosis. Pharmacological inhibition as well as the silencing of three genes coding VGCC, reduced stemness, inducing CSCs apoptosis and inhibiting AKT and ERK phosphorylation	[113]
**ORAI3**	Lung cancer A549, H23	Cisplatin activates AKT leading to increased ORAI3 expression, which, in turn, increase Ca^2+^ entry (SOCE), which favors pAKT and, hence, stem cell markers (Nanog, SOX2)	[59]

## Data Availability

Not applicable.

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
