# Peer review of "Crosstalk between Ca^2+^ Signaling and Cancer Stemness: The Link to Cisplatin Resistance"

_ijms, 2022, doi:10.3390/ijms231810687_

Round 1

Reviewer 1 Report

This manuscript needs an extensive editing about the topics reported. Please, take into consideration the following points before a second evaluation for publication acceptance.

-       At this stage, in the current version of the text, paragraph 2.1 is not useful. That section aims to explain the CDDP mechanism of action, but after few sentences the authors invite the readers to refer to other reviews. Please, delete it or improve the section according to the title.

-       A concise paragraph about calcium signaling inside cells is highly recommended. This paragraph should include players and function of calcium fluxes both inside cells and from the extracellular milieu. Please, refer to these reviews to have an idea (PMID: 35131483, 24291103). Also, mitochondria associated membrane (MAM) compartment should me mentioned and introduced as it plays a pivotal role in cancer and resistance (PMID: 32437958, 29491385).

-       Chapter 2.2 contains a lot of information, but when the reader finishes reading the paragraph, he is not yet clear how calcium fluctuates in the cell and in which manner this is linked to cancer or to resistance or to CDDP. I suggest explaining better this section by highlighting what happens under these conditions in terms of mitochondrial calcium, cytosolic calcium, ER calcium and at the contact sites MAMs. Linking this information to protein expression and function. A figure may help. There are too few figures in this manuscript.

-       If possible, more data or information on the topic from clinical studies should be reported. This may increase the impact of this review. Likewise, also data from preclinical models.

-       Despite the review is focused on cancer stemness, the background about this topic is absent in the manuscript. I suggest explaining in depth what is cancer stemness.

-       It is known a link between Cisplatin and autophagy (PMID: 31885310), autophagy and calcium (PMID: 34440894), autophagy and cancer (PMID: 34830777). What about these topics in CSCs?

-       References are almost dated.

Author Response

#please see the attachment#

Reviewer 2 Report

Reviewer’s comments to “Crosstalk between Ca2+ signaling and cancer stemness: the link to cisplatin resistance” by Kouba et al., ijms-1830459

Cisplatin and other related platinum-containing drugs are widely used in anticancer chemotherapy. Their usefulness is usually limited by the emergence of drug-resistant tumour cell populations, and this may be associated with a phenotype reminiscent of stem cells. A significant set of experimental observations suggests that cisplatin treatment, the presence of cancer stem cells in tumours and cellular calcium homeostasis are interconnected. Understanding the underlying molecular and cell biological mechanisms more fully would be important in order to enhance the antitumour effects of cisplatin and to fight relapse after chemotherapy that initially seems to be efficacious. Making cisplatin-containing chemotherapy regimens more efficient is an important goal of present-day cancer therapy and research. The concept of “cancer stem cells” has undergone important developments, and these cells seem to be involved in drug-resistant relapse. Moreover, a significant amount of experimental data has been accumulated regarding the involvement of calcium signalling in the molecular mechanisms that lead to the acquisition of a stem cell-like phenotype in neoplastic cells.

In this work Authors review the cross-talk between cisplatin treatment and the emergence of cancer stem cells, with special attention to the interactions between cisplatin and various cellular calcium channels involved in cellular calcium signalling and the regulation of “stemness”. Overall, the data reviewed in this work, including also original work by the Authors, show convincingly, that calcium signalling and cisplatin resistance are interconnected, and that the acquisition of stemlike properties of tumour cells is involved in this process. The involvement of various calcium channel types is discussed in detail, and this may allow in the future to propose calcium channel-targeting pharmaceutical approaches for the enhancement of the therapeutic value of cisplatin.

This work will help to strengthen the dialogue between applied cancer research and calcium channel-related basic science, and will serve as a useful review of the current state of our knowledge on calcium channel-related effects of cisplatin and cisplatin resistance. The paper is therefore timely, focused, and will interest a wide range of readers.

Comments:

It would be nice, if Authors could give a more clear-cut definition of cancer stem cells. This is indispensable for non-specialist readers for the understanding of the paper. What is the relationship between cancer stem cells and normal stem cells? Normal stem cells usually constitute the basis of a hierarchical tree of differentiation that, following several steps of cell division, maturation and intermediate phenotypic states give rise to fully differentiated progeny with specialized effector functions. How is this related to cancer stem cells? Are cancer stem cells transformed normal stem cells? What is the relevance of embryonic stem cell markers in tumours? Is the cancer stem cell state fixed, or is it a reversible phenomenon? (See for example : Rehman and O’Brien, Nature 2022; https://doi.org/10.1038/d41586-022-01866-x ; see also lanes 47-48 of the Manuscript in this context.) Are cancer stem cells stable? Are cancer stem cells a minor subpopulation in tumours? Are stem cells in tumours selected or induced by cisplatin? Are tumour stem cells responsible for drug resistance and, importantly, for metastasis formation? Quiescence and self-renewal of cancer stem cells (as in the Abstract) needs also some clarification, because, formally, pure quiescence (i.e.: absence of cell division) is incompatible with self-renewal, as well as with tumour formation.

Moreover, the notion of “cancer stem cells” in the field is a somewhat vaguely defined, still developing idea, that can vary depending on tumour type, experimental system, detection methods or laboratory preferences. It would be useful for non-specialist readers to discuss the notion of “cancer stem cells” in such a way that, in addition to the basic idea and various, albeit converging definitions, the multi-faceted nature of this notion, as well as its limitations can also be understood by readers, because the lack of this may lead to misunderstandings and oversimplifications. In summary, non-specialist readers deserve more extensive guidance in a dedicated paragraph, regarding the important notion of cancer stem cells, in the introductory part of this Paper.

Lane 37 : “The first test results obtained on rat sarcomas in the 1960s supported the role of CDDP as an anti-cancer drug…” :  “rat sarcomas” is a rather vague notion. These experiments were conducted of well-defined rodent sarcoma (and leukæmia) models. It would be nice, if these could be specified (Yoshida- or Jensen rat sarcoma, murine sarcoma 180 or others?)

Probably it would be appropriate to include PMID: 5782119 (Barnett et al. (1969) Nature) as an original publication on platinum drug cytotoxicity.

Lanes 158-159 :  “The activated PLC (Phospholipase C) cascade induces the hydrolysis of inositol biphosphate (PIP2) into Diacylglycerol (DAG) and inositol triphosphate (IP3)” The Reviewer believes that the substrate cleaved by PLC is the phosphoinositide phosphatidylinositol 4,5-bisphosphate (PIP2).

As discussed in this work, cisplatin effects mediated through various calcium channels are probably mainly mediated through calcium. By modifying various calcium permeability parameters or the expression/localization of the target calcium channel, cisplatin will modify cellular calcium homeostasis and signalling, gene expression and organelle function. It would be nice, if, similarly to Fig. 1. for apoptosis induction, Authors could include a Figure on cellular calcium homeostasis and signalling, depicting the localization of various calcium channels and also calcium pumps, the Stim-ORAI complex, endoplasmic reticulum-mitochondrion interactions, endoplasmic reticulum stress responses, the PLC-PIP2-IP3/DAG pathway, PKC, calcineurin, NF-kB, NFAT etc. Calcium-dependent cisplatin sensitization requires also some additional discussion regarding molecular mechanisms.

Cisplatin can covalently modify, in addition to DNA, also proteins (see for example : Fuertes, Curr. Medicinal Chem., 2003). May this have some relevance regarding its effects on ion channels?

Lane 637 :  Ref. 28 “SHEN” : Shen

Lane 742 : Please correct the names of Authors in Ref. 75.

Lane 93 : “p53: transcription factor” : ?

Lane 87 : “checkpoints controls” : “checkpoint controls” or “checkpoint control” or “control checkpoints” ?

Lane 101 : “The expression of Ca2+ channels have been demonstrated…” : has been demonstrated…

Lane 174 :  “know-down” : knock-down

Lane 188 :  “using A549 cell line” : using the A549 cell line

Lanes 192-193 :  “This difference between both studies could be explained by…” : This difference between these two studies could be explained by…

Lanes 195 and elsewhere : “hepatocarcinoma” : given that several different carcinoma types can arise in the liver, “hepatocellular carcinoma” would be more appropriate (as per lane 324; otherwise please specify the specific intrahepatic carcinoma type.)

Lane 197 : “ovary cancer” : ovarian cancer

Lanes 206-207 : “has also reported in an hepatocarcinoma cell line” : has also been reported in a hepatocellular carcinoma cell line

Lane 211 : “might confers” : might confer

Lane 212 : “for prostate cancer cell line” : for a prostate cancer cell line ?

Lane 213 : Please insert a Reference.

Lane 219 and Table 1 : « T-47-D », « T47-D » : T-47D

Lane 224 : « treated-patients » : ?

Lanes 236-237 : “Another study on MCF-7 breast cancer showed the involvement of…” Please note that MCF-7 is not “a breast cancer”, but a breast cancer-derived cell line.

Lane 239 : “by the capsazepine” : by capsazepine

Lane 240 : “limited the increase of CDDP-intracellular Ca2+ concentrations” : ?

Lanes 245-246 : “CDDP-based chemotherapy plays an important role in the treatment of cancers. Unfortunately, the development of resistance has become a major therapeutic challenge” : In the opinion of the Reviewer, cisplatin resistance “has not become” a therapeutic challenge (as, for example, antibiotic resistance has); rather, cisplatin resistance was a problem from the beginning; cancer is not a propagative condition in the human population.

Lanes 247-248 : “One way to explain this resistance is based on emergence or expansion into the cancer cell population of stem cell markers” : in the opinion of the Reviewer, the problem is not the emergence or expansion of those markers, but the emergence or expansion of resistant cells (that carry those markers).

Lanes 252-253 : “and ALDH is known as the detoxification enzyme” : “and ALDH is known as the most important/a main detoxification enzyme?

Lane 254 : “but they exist tumor-specific stem cells markers” : there exist…?

Lanes 274, 275 : “inmate chemoresistance” : innate

Lanes 289-290 : “especially for the patients with chronic renal diseases.” : especially for patients with chronic renal diseases.

Lane 293 : please add “beta”.

Lane 295 : “expansion of ALDH1 positive activity subpopulation” : expansion of ALDH1 activity-positive subpopulation

Lanes 305-307 : “That being said, the hypothesis of aberrant Ca2+ signals contributing to altered CSC function in promoting cancer development and resistance could also represent important targeting against CSCs.” : Please reformulate.

Lanes 320-321 : “Consequently, the hypothesis of channels implicated in this process may also represent crucial targets against CSCs.” : Please reformulate.

Lane 335 : “xenografts mice models” : mouse xenograft models ?

Lane 339 : “Nickel” : nickel

Lane 361 : “might favors” : might favor

Lane 361 : “has net been “ : has not been

Lanes 369-370 : “As for TRP channels, until this day, few members of the TRP family have been linked to stem cell function which are TRPC3, TRPA1, TRPV1, TRPV2 and TRPM7.” : As for TRP channels, until this day, only a few members of the TRP family have been linked to stem cell function, these are TRPC3, TRPA1, TRPV1, TRPV2 and TRPM7.

Lane 379 : “TRPV2 has been directly associated with the self-renewal course of CSCs” : course?

Lane 383 : “tranilast” : please delete quote (“…“).

Lane 391 : “This course was also observed in vivo liver cancer model,” : This course was also observed in an in vivo liver cancer model. (Or: This course was also observed in vivo, in a liver cancer model.)

Lanes 394-395 : “Differentiated stem cells had higher levels of TRPV2 expression…” : what are “differentiated stem cells” ?

Lanes 404-406 : “When CBD treatment was combined with conventional chemotherapy agent, Carmustin, apoptosis of Gliblastoma Stem Cells was enhanced and improved outcomes on tumor progression.” : When CBD treatment was combined with conventional chemotherapy agent Carmustin, apoptosis of Gliblastoma Stem Cells was enhanced and improved outcomes on tumor progression were obtained.

Lanes 412-413 : “As for TRPM7, two research works have also demonstrated TRPM7 expression in glioblastoma and neuroblastoma stem cells respectively…” : As for TRPM7, two research works have also demonstrated TRPM7 expression in both glioblastoma and neuroblastoma stem cells…

Lanes 416-417 : “and ALDH1 and inhibited…” : and ALDH1, and inhibited…

Lane 419 : “SH-SY5Y neuroblast cell line…” : SH-SY5Y neuroblastoma cell line…

Lane 426 : Please mention the pharmacologic activity of waixenicin A (Inhibitor? Activator?)

Lane 439-440 : “PYK2 439 (pyruvate kinase 2)…” : the Reviewer is not sure whether PYK2 is indeed a pyruvate kinase. Isn’t this rather the protein tyrosine kinase 2b enzyme encoded by the PTK2B gene?

Lane 451 : “in vivo…” : In vivo

Lanes 451-452 : “a transgenic mouse model that develop spontaneous metastatic melanoma… : that develops…

Lanes 455-456 : “Ca2+ release from IP3Rs also seem to be involved in quiescence…” : …seems to be involved…

Table 2 (TRPM7) : “leads tumor sphere formation…” : leads to tumor sphere formation…

Table 2 (IP3R) : “Decrease of IP3R expression mediates Ca2+ flux and reduces expression of stem pathways.” : please note that a decrease of IP3R expression probably mediates a decrease of calcium flux…

Lanes 460-461 : “cancer stem cells populations” : cancer stem cell populations

Lane 471 : “than to parental cells” : than parental cells

Lane 475 : “drugs resistance”: drug resistance (or resistance to drugs)

Lanes 477 and 479 : “...neutralized antibody” : ...neutralizing antibody?

Lanes 486-487 : “showed an improved response than the group treated with chemotherapy alone.” : showed an improved response when compared to the group treated with chemotherapy alone

Lane 488 : “CSC-enriched spheres ovarian cancer cell lines” : ?

Lane 494 : “…is upregulated in ovarian CSCs than in ovarian cancer cells.” : is more upregulated…? …is upregulated when compared to…?

Lane 506 : “adenocarcinomas tissues” : adenocarcinoma tissues

Lane 506 : “when compared to adjacent ones” : when compared to adjacent normal tissue ?

Lane 509 : “through AKT pathway” : through the AKT pathway

Lane 510 : “using Immunohistochemical assay” : using an immunohistochemical assay

Table 3 (VGCC) : “Several PDXs models” : Several PDX models

Table 3 (VGCC) : “channel subunits expression...” : channel subunit expression... (or: The expression of the L- and T-type Ca2+ channel subunits is upregulated)

Lane 534 : “It’s” : It is

Lane 531 : via (lane 519) or via ?

Lanes 516-518 : “Cells overexpressing ORAI3 possess resistance to CDDP-induced apoptosis and the ones downregulating ORAI3 were prone to cellular apoptosis particularly due to CDDP exposure.” Please discuss here, whether this is an ORAI3-specific effect. If so, please discuss, what is the mechanistic basis of such an ORAI isoform-specific effect?

Lane 547 : “lin” : Lin

Lanes 547-548 : “They found that p53 is able to bind to the promoter of Nanog and control its expression.” “control” meaning inhibit?

Lanes 552-553 : “The authors demonstrated that the Ca2+ influx mediated by ORAI3 is able to regulate p53 activity and allowing resistance to chemotherapeutic drugs in breast cancer cells.” : The authors demonstrated that the Ca2+ influx mediated by ORAI3 is able to regulate p53 activity and allow (induce?) resistance to chemotherapeutic drugs in breast cancer cells.

Lane 554 : “to Nanog increased expression” : to increased Nanog expression

Lane 559 : “The usage of CDDP…” : The use… ?

Lane 568 : “However, based on the few studies mentioned in this review,” : few ?

Lanes 22-23 : “…if we can understand the molecular mechanisms linking Ca2+ to CDDP-induced resistance and CSCs behaviors, alternative and novel therapeutic strategies could be considered.” : It would be interesting for the readers, if Authors could elaborate on such future possible therapeutic strategies in greater detail in the Discussion.

The Paper requires editing by a native English speaking scientific editor before publication.

It would be nice, if it could be stated more clearly, what makes cisplatin particularly special, when compared to other DNA-damaging agents, regarding cancer stem cell biology and calcium. Intuitively it may not be immediately clear for non-specialists, what mechanisms set platinum drugs apart from other drugs such as, for example, other types of alkylating agents.

Author Response

#Please seen the attachment#

Round 2

Reviewer 1 Report

The authors minimally accepted my suggestions aimed to expand and improve the manuscript. For this reason I have to reject the manuscript.

Reviewer 2 Report

Reviewer’s comments to the revised manuscript

Authors addressed successfully most issues raised by the Reviewer. New Figures have been included, as well as an enhanced introduction regarding the notion of cancer stem cells. The following issues need to be addressed prior publication. These are related to the Author’s Reply and to new material added to the revised Manuscript :

11)     Following a question raised by the Reviewer, Authors state in the Accompanying Letter, that “To our knowledge interactions between ion channels and cisplatin has not been widely explored in the literature. However, many studies focused on the lipid membrane-platinum interactions. It has been well established that cisplatin interacts with plasma membrane phospholipids that leads to changes in the biophysical properties of the membranes (e.g., fluidity and permeability). We assume that those changes may interfere with the activity of some ion channels that are sensitive to lipids, which may affect sensitivity and resistance phenotypes of cells to platinum complexes.” Given the pertinence of the issue to this Paper, it would be nice if Authors could incorporate the above text, or a shortened version of it, in the Manuscript.

22)     Lane 389 : The Reviewer suggest replacing “that being said” by “thus”, “consequently” or something similar.

33)     Lanes 484-485 : “Higher TRPV2 expression in differentiated cells lead to the increase of basal Ca2+ levels”  : “leads”

44)     Lanes 520-521 : “When TRPM7 was treated with an inhibitor Waixenicin A, lung cancer cells failed to form tumorspheres…” : When TRPM7 was treated with an inhibitor, Waixenicin A, lung cancer cells failed to form tumorspheres…” or “When TRPM7 was treated with the inhibitor Waixenicin A, lung cancer cells failed to form tumorspheres…”

55)     Lane 668 : “CDDP exposure affects the expression of Ca2+ channels and proteins.” : Please note that these calcium channels are themselves proteins. “CDDP exposure affects the expression of Ca2+ channels and other calcium homeostatic proteins.” ?

66)     In Acknowledgements the use of uppercase in the names of organizations would probably require some further attention.

77)     Lane 694 : “Nanocarriers” : nanocarriers ?

88)     Lanes 689-694 : “At this point, gene-targeted therapy is an alternative tool for calcium channel blockers that is being more applied to treat Ca2+ channel-related cardiovascular diseases. Similarly, mRNA-based therapies that bind to Ca2+ channels’ mRNA leading to their degradation is also being explored. However, these approaches in some way must be validated by a pharmacological approach, therefore, other pharmacological alternatives must be taken into account such as the development of Nanocarriers to improve both ion channel inhibitors and CDDP bioavailability.” : In the opinion of the Reviewer, this is a rather general and somewhat vague way of discussing this issue. In addition, it is not clear, how gene therapies, nanocarriers etc., may be efficacious in cancer, considering that these seldom attain the entire tumor stem cell population, leading to regrowth of unaffected tumor cells.

Please discuss, the function of exactly which channels, and in which tumor types, should be inhibited (or enhanced) pharmacologically or by other means, in order to obtain, potentially, an effect against tumor stem-cells.

99)     For lanes 516-518 in the original Manuscript, Authors state in their Response that “The effect that CDDP increases the expression of some calcium channels is not specific to ORAI3, it is also observed for ORAI1 and STIM1. We suggest that CDDP, by activating PI3K/Akt, lead to increase the expression of both ORAI1 and ORAI3 in ovarian and breast resistant cancer cells respectively.” : Please include this in the Manuscript. (And : “leads to an increased expression of…”)

110)  In their Reply, Authors state that “We agree with the reviewer that it would be very important to compare the role of calcium signature of cancer stem cells in CDDP-resistance with the others alkylating agents, but given the short time attributed by the editor, it has been a bit challenging to the authors to add and merge this topic with this review.”

The Reviewer understands the time constraint. However, given the subject of the Manuscript, the fact that cisplatin belongs to a larger family of alkylating agents which may also have effects on calcium homeostasis (see for example, PMID: 27738742, Stenger et al., Arch. Toxicol., 2017) clearly needs to be briefly mentioned and cited, because it is possible, importantly, that the anticancer effects of these (non-cisplatin) alkylating cytostatics may also be enhanced by manipulating calcium channels or the calcium homeostasis of cancer stem cells in general.

111)  The Paper is centered on cisplatin. Are there data on other platinum drugs such as oxaliplatin or carboplatin?
